# Evidence of Phosphate Mining and Agriculture Influence on Concentrations, Forms, and Ratios of Nitrogen and Phosphorus in a Florida River

**Shuiwang Duan [1],\*, Kamaljit Banger [2] and Gurpal S. Toor [1]**

[1] Department of Environmental Science and Technology, University of Maryland, College Park, MD 20742, USA; gstoor@umd.edu
[2] School of Environmental Sciences, University of Guelph, Guelph, ON N1G 2W1, Canada; kamal01@uoguelph.ca
\* Correspondence: sduan@umd.edu; Tel.: +1-(301)-405-2851

**Abstract:** Florida has a long history of phosphate-mining, but less is known about how mining affects nutrient exports to coastal waters. Here, we investigated the transport of inorganic and organic forms of nitrogen (N) and phosphorus (P) over 23 sampling events during a wet season (June–September) in primary tributaries and mainstem of Alafia River that drains into the Tampa Bay Estuary. Results showed that a tributary draining the largest phosphate-mining area (South Prong) had less flashy peaks, and nutrients were more evenly exported relative to an adjacent tributary (North Prong), highlighting the effectiveness of the mining reclamation on stream hydrology. Tributaries draining > 10% phosphate-mining area had significantly higher specific conductance (SC), pH, dissolved reactive P (DRP), and total P (TP) than tributaries without phosphate-mining. Further, mean SC, pH, and particulate reactive P were positively correlated with the percent phosphate-mining area. As phosphate-mining occurred in the upper part of the watershed, the SC, pH, DRP, and TP concentrations increased downstream along the mainstem. For example, the upper watershed contributed 91% of TP compared to 59% water discharge to the Alafia River. In contrast to P, the highest concentrations of total N (TN), especially nitrate + nitrite ($NO_x$–N) occurred in agricultural tributaries, where the mean $NO_x$–N was positively correlated with the percent agricultural land. Dissolved organic N was dominant in all streamwaters and showed minor variability across sites. As a result of N depletion and P enrichment, the phosphate-mining tributaries had significantly lower molar ratios of TN:TP and $NO_x$–N:DRP than other tributaries. Bi-weekly monitoring data showed consistent increases in SC and DRP and a decrease in $NO_x$–N at the South Prong tributary (highest phosphate-mining area) throughout the wet season, and different responses of dissolved inorganic nutrients (negative) and particulate nutrients (positive) to water discharge. We conclude that (1) watersheds with active and reclaimed phosphate-mining and agriculture lands are important sources of streamwater P and N, respectively, and (2) elevated P inputs from the phosphate-mining areas altered the N:P ratios in streamwaters of the Alafia River.

**Keywords:** phosphate-mining; mine reclamation; agriculture; nitrogen; phosphorus; Alafia River; Florida

## 1. Introduction

Human alterations of the global nitrogen (N) and phosphorus (P) cycles have increased coastal eutrophication resulting in harmful algal blooms and changes in aquatic food webs [1,2]. In the eutrophicated coastal water bodies such as Tampa Bay Estuary [3,4], nonpoint sources are the dominant contributors of nutrients, which are often difficult to control due to the diversity of spatial areas that contribute different nutrient sources within watersheds [5–7]. In general, human-dominated land-uses (urban and agriculture), which receive external nutrient inputs from fertilizers, wastewater, septic systems, are often associated with higher nutrient loadings and different nutrient forms relative to natural areas [8–10]. High nutrient inputs from human-dominated land uses can also be attributed

to altered delivery processes to water bodies. For example, impervious surfaces with urbanization increase the magnitude of stormwater runoff and nutrient exports, which minimizes the removal of reactive N and P, resulting in higher exports of nutrients [11–13].

Phosphorus fertilizers from phosphate-mining are one of the major P sources of coastal eutrophication, but surprisingly less is known on how phosphate-mining affects nutrient export to receiving waters relative to other land-use activities [14]. Phosphate-mining has a long history in Florida, United States [15]. The central Florida region known as Bone Valley is still considered one of the most economically accessible phosphate deposits in the world [16]. There are 27 phosphate mines in Florida, covering more than 1820 km$^2$ or 1% of the state [17]. Today, nine phosphate mines are active, most of which are in central Florida. Because runoff from mining operations may influence the hydrology, water quality, and habitat loss, the Florida Legislature requires that all lands mined for phosphate after 1 July 1975 be reclaimed [15]. A recent examination on water quality changes in two watersheds of central Florida showed consistent decreases in total P (TP), but TP criteria violations were still severe [14]. Little is known about the forms and sources of P in phosphate-mining watersheds, which is essential for further improvement for the mining reclamation.

The objective of this research was to examine if phosphate-mining with reclamation in Florida affects nutrient abundance, forms, and ratios, compared to other types of land uses. Our guiding hypothesis was that P losses are greater in the phosphate-mining than the human-impacted (agricultural and urban) or the forested sub-basins despite recent mining reclamation actions. We also hypothesized that the N to P ratio and forms of P loss from the phosphate-mining sub-basins are different from that of other sub-basins, with lower N:P ratios and more reactive P in mining sub-basins due to the release of reactive P from the mining sites. To test this hypothesis, we selected Alafia River Watershed (ARW), a mixed land-use watershed in central Florida with a large phosphate-mining area [14,18]. We investigated the transport of inorganic and organic forms of N and P in streamwaters of different sub-basins during a wet season that receives 50 to 70% of annual rainfall in four months (June–September). This study provides a comprehensive evaluation of the transport of different N and P forms from diverse land use in a watershed, which will help design and fine-tune effective nutrient pollution remediation programs for coastal waters like Tampa Bay.

## 2. Materials and Methods

### 2.1. Study Site

The ARW is located in west-central Florida and drains 1085 km$^2$ of land to the Tampa Bay Estuary (Figure 1). The soils in the watershed are sandy, with moderate to slow infiltration, and are dominated by Myakka, Winder, Zolfo, Lake, and Chandler soil groups [19]. The climate at the study site is subtropical, with a mean long-term annual precipitation of 118 cm, 63% of which (74 cm) occurs during the wet season (June–September). In 2009, annual precipitation was 116 cm, 53% of which occurred during the wet season (Figure 2).

Three mainstem stations (Alafia, Bell Shoals, and Lithia; hereafter referred to as M1, M2, and M3, respectively) drain 80–99% of the watershed. Lithia (M3) station receives flow from North Prong, South Prong, and English Creek, whereas Bell Shoals (M2) station receives flow from Turkey Creek and Fishhawk Creek in addition to M3 station (Figure 1). The most downstream station, Alafia (M1), receives the discharge from all the sub-basins and drains approximately 99% of the watershed. Based on the proportion of land use (Table 1), we classified sub-basins into four types: (i) phosphate-mining (>36%; South Prong and North Prong), (ii) agricultural (30%; Turkey Creek), (iii) urban (66%; Buckhorn Creek), and (iv) forest (>32%; Fishhawk Creek and Bell Creek). One sub-basin, English Creek, is between urban and agricultural types due to similar urban and agricultural land use (27% vs. 22%). Among the two phosphate-mining sub-basins, North Prong (NP) drains much more urban land (24%) than South Prong (SP) (4%). Additionally, the agricultural sub-basins of Turkey Creek and English Creek have 19% and 10% of phosphate-mining land, respectively.

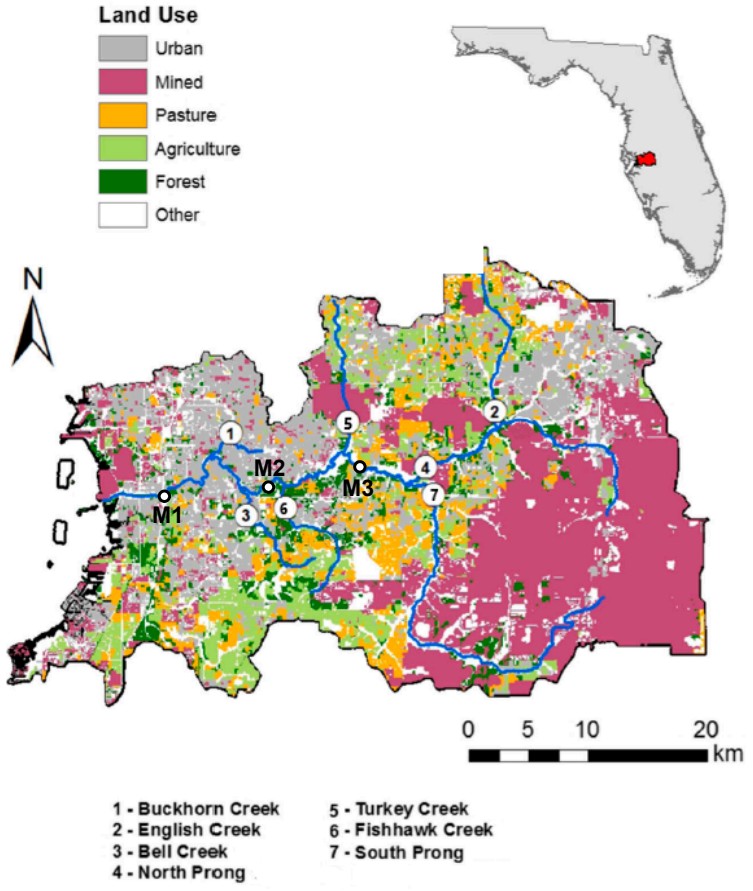

**Figure 1.** A location map showing different sub-basins of the Alafia River Watershed located in central Florida, United States. M1 refers to Alafia, M2 refers to Bell Shoals, and M3 refers to Lithia mainstem stations.

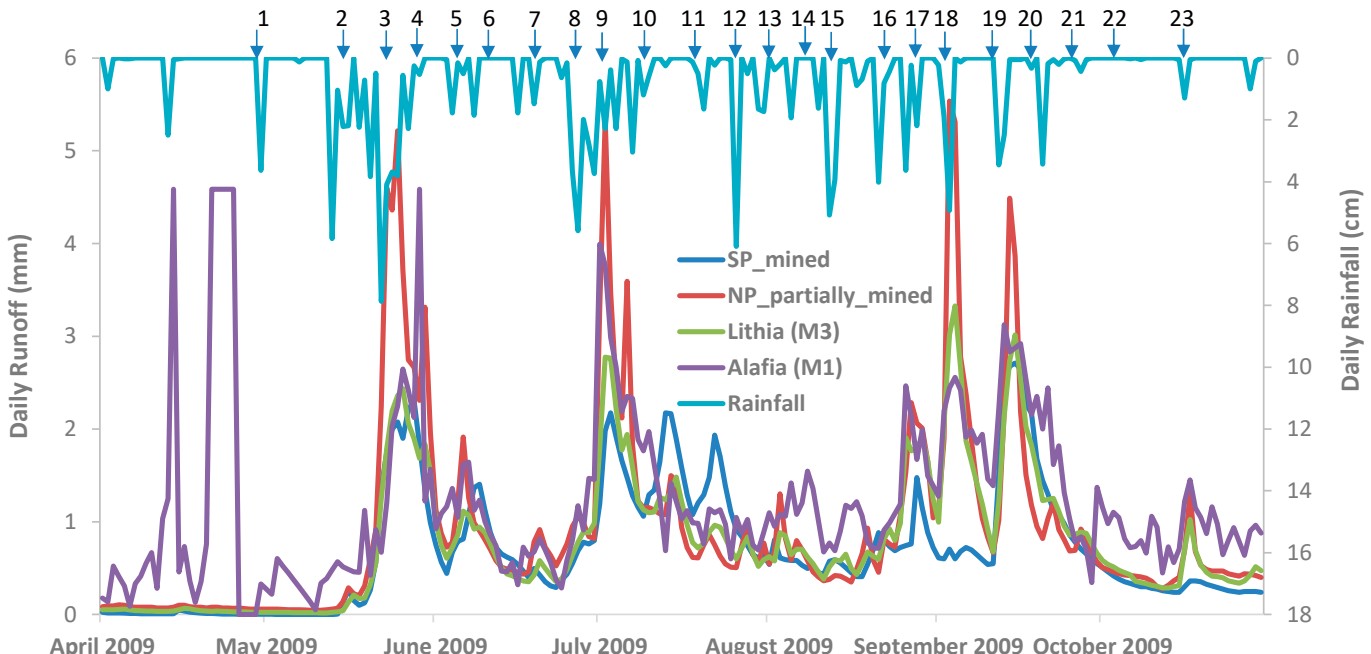

**Figure 2.** Daily rainfall and daily stream runoff for selected tributaries of the Alafia Watershed from April to October 2009. SP and NP are abbreviated for South Prong and North Prong, respectively. The labels from 1 to 23 on top of the figure indicate the timing of the 23 sampling events.

**Table 1.** Characteristics and land use in the Alafia River Watershed sub-basins located in central Florida, United States.

| Sub-Basin | Type | Lati-Tude | Longi-Tude | Drainage Area (km²) | Septic Systems [1] | Land-Use in 2009 (% of Total Sub-Basin) [2] | | | | |
|---|---|---|---|---|---|---|---|---|---|---|
| | | | | | | Urban | Agriculture | Wetland | Forest | Mined |
| Lithia (M3) | Mainstem | 27.86 | −82.16 | 860 | - | 19 | 13 | 13 | 6 | 39 |
| Bell Shoals (M2) | Mainstem | 27.86 | −82.26 | 974 | - | 19 | 13 | 13 | 6 | 36 |
| Alafia (M1) | Mainstem | 27.87 | −82.32 | 1072 | - | 20 | 12 | 13 | 6 | 35 |
| South Prong | Trib._mined | 27.86 | −82.13 | 277 | 0.01 | 4 | 11 | 13 | 4 | 59 |
| North Prong | Trib_mined | 27.86 | −82.13 | 350 | 0.57 | 24 | 10 | 13 | 16 | 36 |
| Turkey Creek | Trib_agr. | 27.91 | −82.18 | 128 | 0.11 | 25 | 30 | 10 | 4 | 19 |
| English Creek | Trib_agr/urban | 27.93 | −82.06 | 99 | 0.10 | 27 | 22 | 14 | 6 | 10 |
| Buckhorn Creek | Trib_urban | 27.90 | −82.17 | 19 | 0.37 | 66 | 10 | 8 | 12 | 0 |
| Bell Creek | Trib_forest | 27.80 | −82.18 | 90 | 0.12 | 26 | 12 | 10 | 37 | 1 |
| Fishhawk Creek | Trib_forest | 27.85 | −82.24 | 70.6 | 0.007 | 14 | 12 | 5 | 32 | 0 |

[1] Number of septic systems per hectare (data from [18]). [2] The sum of the total land use is not 100% since other land-uses such as shrublands, brushlands, and open lands are not included. There was one wastewater treatment plant each in Turkey Creek and North Prong. The information of drainage area and land use was calculated from the land use map (Figure 1), which was derived from NLCD_2011_Land_Cover, utilizing the Florida Department of Transportation Land Use, Cover and Forms Classification (FLUCCS) System. Tributary is abbreviated as Trib.

## 2.2. Streamwater Sampling

Ten sites (3-mainstem and 7-tributaries, Figure 1) were sampled over 23 sampling events during April–October 2009 (Figure 2). The frequency of sample collection was bi-weekly from April to mid-May 2009 ($n = 2$), followed by weekly sampling from May to September 2009 ($n = 18$), and bi-weekly in October 2009 ($n = 3$). During each sampling event, specific conductance (SC), dissolved oxygen (DO), and streamwater pH were measured in-situ using Manta 2 Water Quality Multiprobe (Eureka, Austin, TX, USA), and a grab water sample was collected from the channel thalweg (center of the streamflow). Before collecting grab samples, the water bottle was rinsed three times with the streamwater, and 250 mL of the water sample was collected. The collected samples were chilled on ice and transported to the laboratory.

Approximately 50 mL of the sample was filtered within 24 h of collection using a 0.45 µm filter (Pall Corporation, Ann Arbor, MI, USA). The filtrates were preserved in 20 mL plastic scintillation vials with 2-drops of sulfuric acid ($H_2SO_4$) to prevent biological activity and refrigerated at 4 °C. All the samples were analyzed within 28 days of collection.

## 2.3. Streamwater Chemical Analysis

A filtered subsample was analyzed for $NH_4$–N and $NO_x$–N ($NO_2$–N + $NO_3$–N) using a Discrete Analyzer (Model AQ2, Seal Analytical, Mequon, WI, USA) with EPA methods 350.1 [20] and 353.2 [21], respectively. The filtered samples were also analyzed for chloride ($Cl^-$) using the Discrete Analyzer with EPA method 325.2 [22]. Unfiltered and filtered water samples were analyzed for total N (TN) and total dissolved N (TDN), respectively, using the alkaline persulfate digestion method [23] followed by $NO_x$–N analysis as described above. Dissolved organic N (DON) was calculated by subtracting dissolved inorganic N ($NO_x$–N plus $NH_4$–N) from TDN. Similarly, particulate organic N (PON) was calculated by subtracting TDN from TN.

All water samples were analyzed for dissolved reactive P (DRP) and total dissolved P (TDP) in filtered samples and total reactive P (TRP) and TP in unfiltered samples using a Discrete Analyzer (Model AQ2, Seal Analytical, Mequon, WI) with EPA Method 365.1 [24]. For TDP and TP analyses, samples were first digested using persulfate digestion [23], which converts organic P and polyphosphates into orthophosphate. Other P forms were calculated as follows: dissolved unreactive P (DUP) = TDP − DRP, total particulate P (TPP) = TP − TDP, particulate reactive P (PRP) = TRP − DRP, and particulate unreactive P (PUP) = TUP − DUP. When the calculated values were negative, they were under the detection limit and were assumed to be zero [7].

The detection limits were 0.004 mg L$^{-1}$ for $NH_4$–N and 0.003 mg L$^{-1}$ for $NO_x$–N, TDN, and TN analyses. The detection limits were 0.003 mg L$^{-1}$ for TDP and TP and 0.002 mg L$^{-1}$ for DRP and TRP analyses. In the laboratory, quality control (second standard), sample duplicate, reagent blank, laboratory fortified blank (LFB), continued control verification (CCV), and spikes were adopted to inspect quality assurance and quality control of analysis.

*2.4. Data Collections and Statistical Analyses*

Water discharge data were collected from US Geological Survey Stations located in the ARW [25]. The data were only available for four sites: Alafia River at Keysville (02301000) for NP, Alafia River near Lithia (02301300) for SP, Alafia River at Lithia (02301500) for M3, and Alafia River near US 301 at Riverview (02301718) for M1. Daily runoff for the four sites was calculated from US Geological Survey flow data by normalizing drainage areas. Daily precipitation data were collected from the nearest National Oceanic and Atmospheric Administration (NOAA) station (ID: GHCND: USC00087205)—Plant City, Florida (28.02084°, −82.13855°), which is located within the watershed [26].

U.S. Geological Survey LOADEST (Load Estimator) model was used for estimating daily N and P fluxes for the above four sites with discrete nutrient concentration measurements and continuous average daily flow. Detail description of the LOADEST model is also available in [27]. Briefly, the measurements of nutrient concentrations and corresponding streamflow for each station were used for model calibrations. An automated model selection option was used to choose the best model from the set of predefined models to obtain the lowest value of the Akaike Information Criterion. LOADEST was run with a daily time step for the study period, and daily estimates from 1 May to 31 October were summarized to produce nutrient loads. Nutrient cumulative export curves of TN and TP were also estimated for each site, using the method described by Duan et al. [28]. Cumulative export curves show the percent of nutrient mass exported with time. To characterize cumulative export, cumulative TN or TP loads were computed by adding their daily values day-by-day from the highest to the lowest.

The mean, median, and standard errors of N and P forms were calculated using Microsoft Excel. Differences in N and P forms in streamwater among sub-basins were tested using Dunnett's post hoc test for analysis of variance (ANOVA). Relationships between N and P forms and land use (or runoff or water quality variables) were tested using Spearman's correlation with $p < 0.05$.

## 3. Results

*3.1. Relationship between Rainfall and Surface Runoff*

According to the records at Plant City, Florida, rainfall occurred during the wet period from 13 May and 20 September (Figure 2). Two rainfall events with high intensity and long duration occurred during 13–27 May and 27 June–9 July. Surface runoff at all the four sites had peak values during 19–30 May and 29 June–13 July, corresponding to the two rainfall events. A third runoff peak occurred during 26 August–23 September, although there was no corresponding heavy rainfall. Stream runoff also varied among the four sites, with more flashy peaks at NP and more flat peaks at SP (Figure 2). Unexpected runoff peaks occurred at site M1 during April dry period, probably due to interference of flow measurements by tides.

*3.2. Nutrient Export*

Export of TN and TP in the Alafia River at the most downstream site (M1) was 683 and 266 metric tons, respectively, during the study period (1 May to 31 October 2009), with a molar TN/TP ratio of 5.7:1 (Table 2). The watershed above the M3 site contributed approximately 60% of water, less TN (approximately 49%) but more TP (approximately 91%) to the Alafia River. Similarly, SP and NP contributed 17% and 28% of water, relatively less TN but more TP to the Alafia River (Table 2). All three sites (M3, SP, and NP) had lower molar ratios of TN:TP loads (<3.4:1).

Water flow duration curves displayed a marked variation among the sites (Figure 3a). In the NP, relatively more water was exported during periods with high flows relative to SP, M3, and M1. For example, 59% of water was exported from NP during the first 25% of the time with higher flows, while the corresponding percentages of water were 53–54% for SP and M3 and 47% for M1. The TN and TP load duration curves at the four sites generally followed the curves of water duration. However, the differences among the sites were

larger for TN but less for TP (Figure 3b,c). For instance, for NP, SP, M3, and M1 sites, the percentages of TN exported during 25% of the time with high flows were 59%, 54%, 50%, and 43%, whereas the percentages of TP were 55%, 55%, 51%, and 47%, respectively.

**Table 2.** Water discharge, loads of total nitrogen (TN) and total phosphorus (TP), and molar ratio of TN:TP in the South Prong, North Prong, mainstem site M3 at Lithia, and mainstem site M1 at Alafia with decreasing phosphate-mining areas.

| Sites | Water ($10^6$ m$^3$) | TN (Metric Ton) | TP (Metric Ton) | TN:TP (Molar) |
|---|---|---|---|---|
| South Prong | 41 (17%) | 86 (12%) | 56 (21%) | 3.4:1 |
| North Prong | 68 (28%) | 182 (27%) | 143 (54%) | 2.8:1 |
| Lithia (M3) | 142 (59%) | 333 (49%) | 243 (91%) | 3.0:1 |
| Alafia (M1) | 241 | 683 | 266 | 5.7:1 |

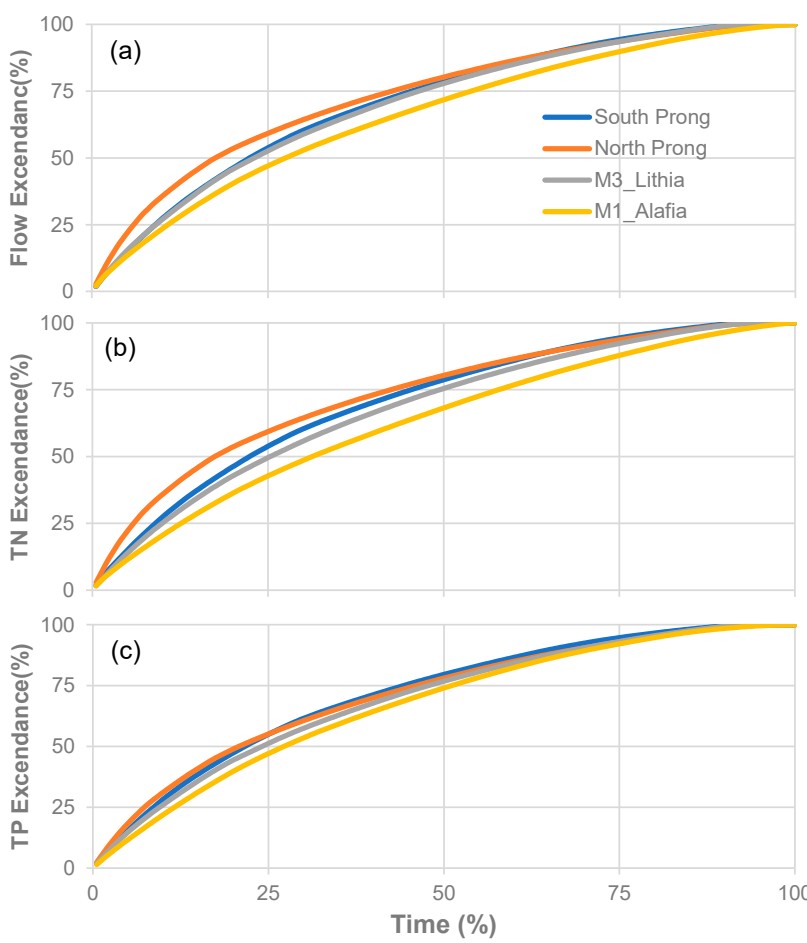

**Figure 3.** Cumulative (**a**) water flow duration, (**b**) total nitrogen (TN), and (**c**) total phosphorus (TP) export as a function of time in South Prong, North Prong, and mainstem sites M3 and M1 of the Alafia River Watershed.

### 3.3. Changes in Streamwater Chemistry across Land Uses

During the 23 sampling events from April to October 2009, the two sub-basins draining more phosphate-mining areas (SP and NP) had significantly higher SC ($0.47 \pm 0.03$–$0.50 \pm 0.03$ µS cm$^{-1}$) and pH ($7.42 \pm 0.09$–$7.46 \pm 0.09$) than the other sub-basins (SC: $0.18 \pm 0.01$–$0.34 \pm 0.03$ µS cm$^{-1}$; pH: $6.74 \pm 0.11$–$7.26 \pm 0.15$) ($p < 0.05$, one-way ANOVA; Figure 4a,c). The lowest values of both SC and pH occurred in the two forested sub-basins (BELL and FS). SC, Cl$^-$, pH, and DO across three mainstem sites were statistically similar, with significantly higher SC ($4.48 \pm 1.58$ µS cm$^{-1}$) and Cl$^-$ ($800 \pm 192$ mg L$^{-1}$) in M1 likely due to tidal influence

(Figure 4a–d). Mean SC and pH were positively correlated with percent of phosphate-mining area when M1 was excluded ($R^2 > 0.88$, $n = 9$–10, $p < 0.01$; Table 3). Different from SC or pH, the concentrations of $Cl^−$ or DO in the two phosphate-mining sub-basins were not significantly different than other sites ($p > 0.05$), although the highest $Cl^−$ concentration was observed in one of the mined sites (NP) (Figure 4b,d).

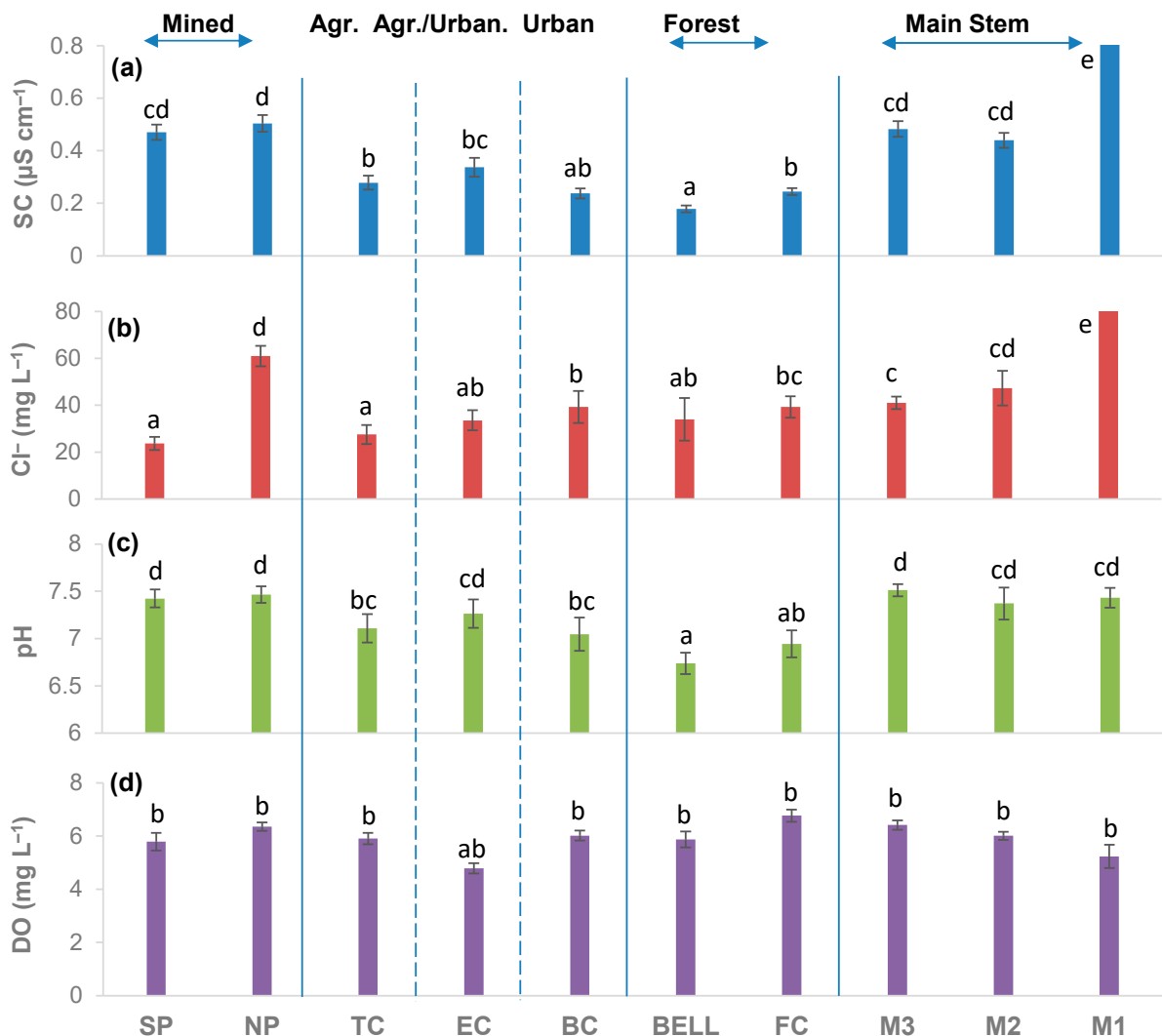

**Figure 4.** Changes in (**a**) specific conductance (SC), (**b**) chloride ($Cl^−$), (**c**) pH, and (**d**) dissolved oxygen (DO) across a land-use gradient of the Alafia River. Stations marked by different letters (a–e) are significantly different at $p < 0.05$. NP: North Prong; SP: South Prong; TC: Turkey Creek; EC: English Creek; BC: Buckhorn Creek; FC: Fishhawk Creek. Note that extremely high values of SC and $Cl^−$ at site M1 ($4.48 \pm 1.58$ $\mu S$ $cm^{−1}$ and $800 \pm 192$ mg $L^{−1}$) are not shown due to limited *y*-axis scales.

**Table 3.** Correlation coefficients between land-use and nitrogen and phosphorus forms in the streamwaters of the Alafia River Watershed over 23 sampling events (April–October 2009).

| Land Use | SC | pH | $NO_x$ | PON | DRP | DUP | PRP | DIN/DRP | TN/TP |
|---|---|---|---|---|---|---|---|---|---|
| Urban | | | | | | | | 0.91 | 0.86 |
| Agricultural | | | 0.82 | 0.55 | | | | | |
| Wetland | 0.73 | 0.75 | | | 0.70 | 0.76 | | | −0.58 |
| Forest | −0.60 | −0.77 | −0.68 | | | −0.65 | −0.61 | | |
| P-mined | 0.90 | 0.85 | | | | | 0.64 | −0.55 | −0.60 |

Values $\geq 0.64$ indicate significant relationships ($p < 0.05$), while the others ($< 0.64$) refer to marginally significant level ($p < 0.1$).

*3.4. Changes in Concentrations, Forms, and Ratios of N and P in Streamwaters*

Among the sub-basins, TN was highest in an agricultural site ($2.95 \pm 0.52$ mg L$^{-1}$) and lowest in two forested sites ($1.99 \pm 0.55$–$2.02 \pm 0.60$ mg L$^{-1}$), whereas TN in two phosphate-mining sub-basins was not significantly different from the others ($p > 0.05$, one-way ANOVA) (Figure 5a). On the other hand, TP in the sub-basins showed a larger variability ($0.51 \pm 0.37$–$2.40 \pm 0.51$ mg L$^{-1}$). TP was significantly higher in four sub-basins ($1.33 \pm 0.41$–$2.40 \pm 0.51$ mg L$^{-1}$) in the upper watershed with phosphate-mining area >10% (two mined sub-basins SP and NP, TC and EC) than the rest of the sub-basins without phosphate-mining ($0.51 \pm 0.37$–$0.71 \pm 0.26$ mg L$^{-1}$) (Figure 5d). As a result of the elevated TP concentrations, TN:TP molar ratio was lower in the four sub-basins in the upper watershed ($2.6 \pm 0.2$–$4.9 \pm 0.3$) than the others ($7.2 \pm 0.9$–$19.7 \pm 2.1$) (Figure 5g). At three mainstem stations, TP concentrations decreased from M3 to M1 site ($1.71 \pm 0.04$ to $1.13 \pm 0.06$ mg L$^{-1}$), leading to an increase in TN:TP molar ratios ($3.2 \pm 0.1$ to $6.0 \pm 0.5$; Figure 5d,g).

Organic N (DON + PON) was the dominant form (1.7–2.2 mg L$^{-1}$; 71–87%) while NH$_4$–N was a minor form (0.10–0.18 mg L$^{-1}$; 4–6% of TN) at all sites, although the two agricultural sites (TC and EC) had more NO$_x$–N (26–31%) than the others (9–19%) (Figure 5b,c). Among N forms, NO$_x$–N and PON displayed larger variability across sites (coefficient of variation or CV: 46% and 25%) relative to DON and NH$_4$–N (CV: 10% and 20%). The highest values of NO$_x$–N and PON occurred in two agricultural sites and, and, the lowest values were in two forest sites (Figure 5b). Mean NO$_x$–N concentration was positively correlated with agricultural land use and negatively with forested land use (Table 3).

In contrast to the dominance of organic N forms, inorganic P forms (DRP + PRP) were dominant (55–79% of TP) while DUP was a minor form (8–14% of TP) at all sampling sites (Figure 5f). Among the sub-basins, DRP and DUP concentrations were significantly higher in the streams draining the four sub-basins with phosphate-mining area >10% (SP, NP, TC and EC) as compared to sub-basins without mining ($p < 0.05$, one-way ANOVA), but PRP and PUP were not significantly ($p > 0.05$) different (Figure 5e). Mean PRP concentration was positively correlated with percent phosphate-mining, while mean DRP and DUP were more closely correlated with percent wetlands area (Table 3).

The molar ratios of dissolved inorganic N (DIN = NO$_x$–N + NH$_4$–N) to DRP were in the range of $0.8 \pm 0.1$–$7.0 \pm 2.0$, and the values were all lower than the corresponding TN:TP molar ratio across sites (Figure 5g). The DIN:DRP molar ratio was lowest in the two phosphate-mining sub-basins (SP and NP; $0.8 \pm 0.1$–$1.2 \pm 0.2$) and was the highest in the urban sub-basins (BC; $7.0 \pm 2.0$). Both DIN:DRP and TN:TP molar ratios were positively correlated with percent urban land use ($p < 0.05$), and negatively correlated with percent phosphate-mining at a marginally significant level ($p < 0.1$) (Table 3).

*3.5. Temporal Variability of Water Quality Variables and Nutrient Concentrations in Selected Sub-Basins*

During the 23 sampling events from April to October, SC was negatively correlated with stream runoff only at the M3 site (R = $-0.65$, $p < 0.05$; Figure 6a–d), whereas Cl$^-$ was negatively correlated with stream runoff at all the four sites (R = $-0.50$ to $-0.66$, $p < 0.05$) (Figure 6e–h). The pH values were also negatively correlated with runoff at the NP and the SP (R = $-0.42$ to $-0.53$, $p < 0.05$) (Figure 6i–l). Although DO varied inversely to stream runoff, the relationship was generally not significant (data not shown).

Different N or P forms showed different temporal variations across stream runoff. In general, dissolved inorganic forms (i.e., NO$_x$–N and DRP) varied inversely to stream runoff (Figure 6m–p,u–x) while particulate nutrients (i.e., PON and TPP) varied similarly to stream runoff (Figure 6q–t,y–ab), however, not all the correlations were significant. Meanwhile, variations of NH$_4$–N, dissolved organic forms (DON and DUP) with stream runoff were not apparent (data not shown).

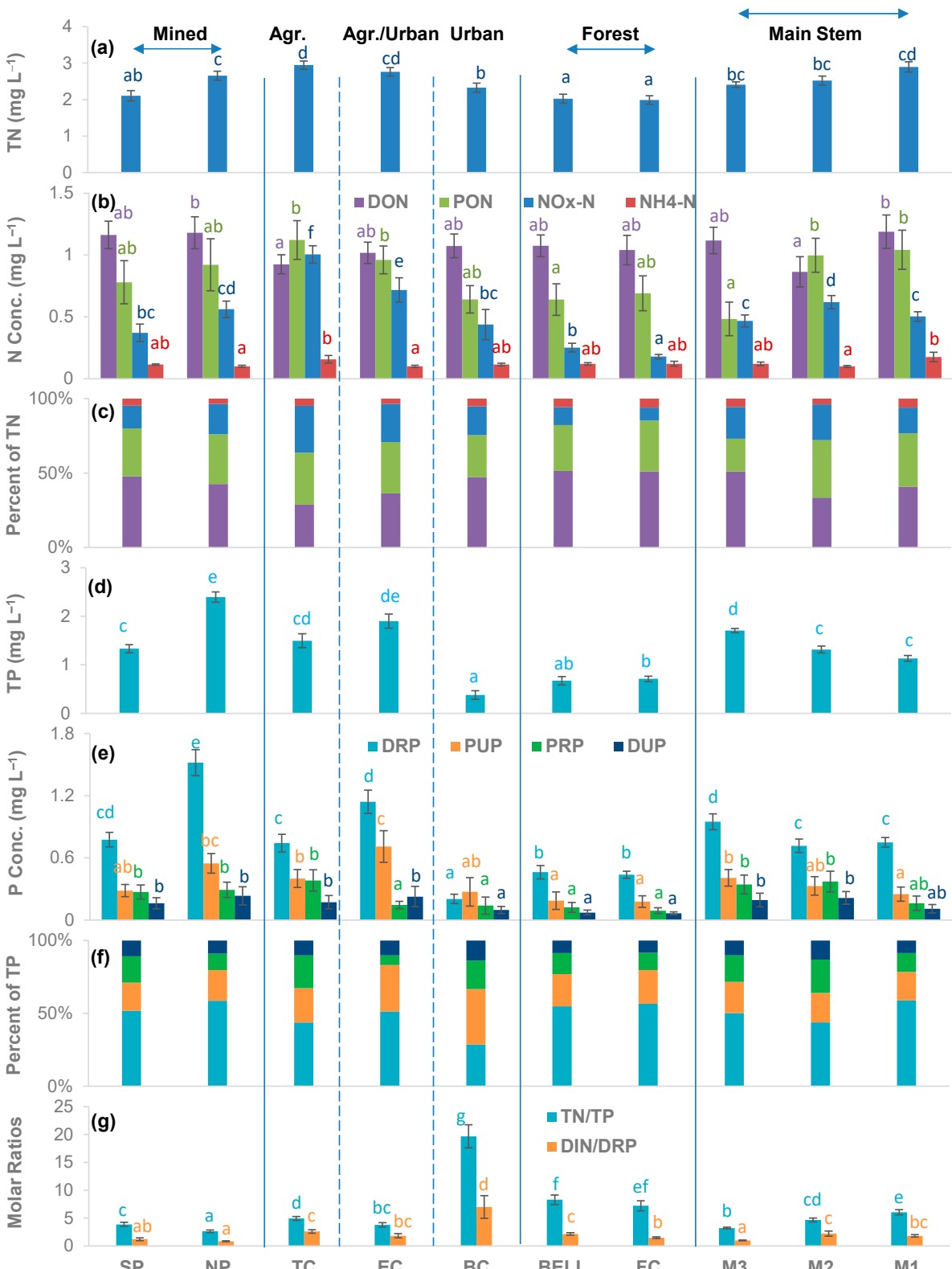

**Figure 5.** Changes in (**a**) concentrations of total N, (**b**) concentrations of N forms, (**c**) proportion of N forms, (**d**) concentrations of total P, (**e**) concentrations of P forms, (**f**) proportion of P forms, and (**g**) molar N to P ratios across a land-use gradient in the Alafia River. Stations marked by different letters are significantly different at $p < 0.05$. NP: North Prong; SP: South Prong; TC: Turkey Creek; EC: English Creek; BC: Buckhorn Creek; FC: Fishhawk Creek. DON: Dissolved organic N; PON: Particulate organic N; DRP: dissolved reactive P; PRP: particulate reactive P; DOP: dissolved organic P; PUP: particulate unreactive P.

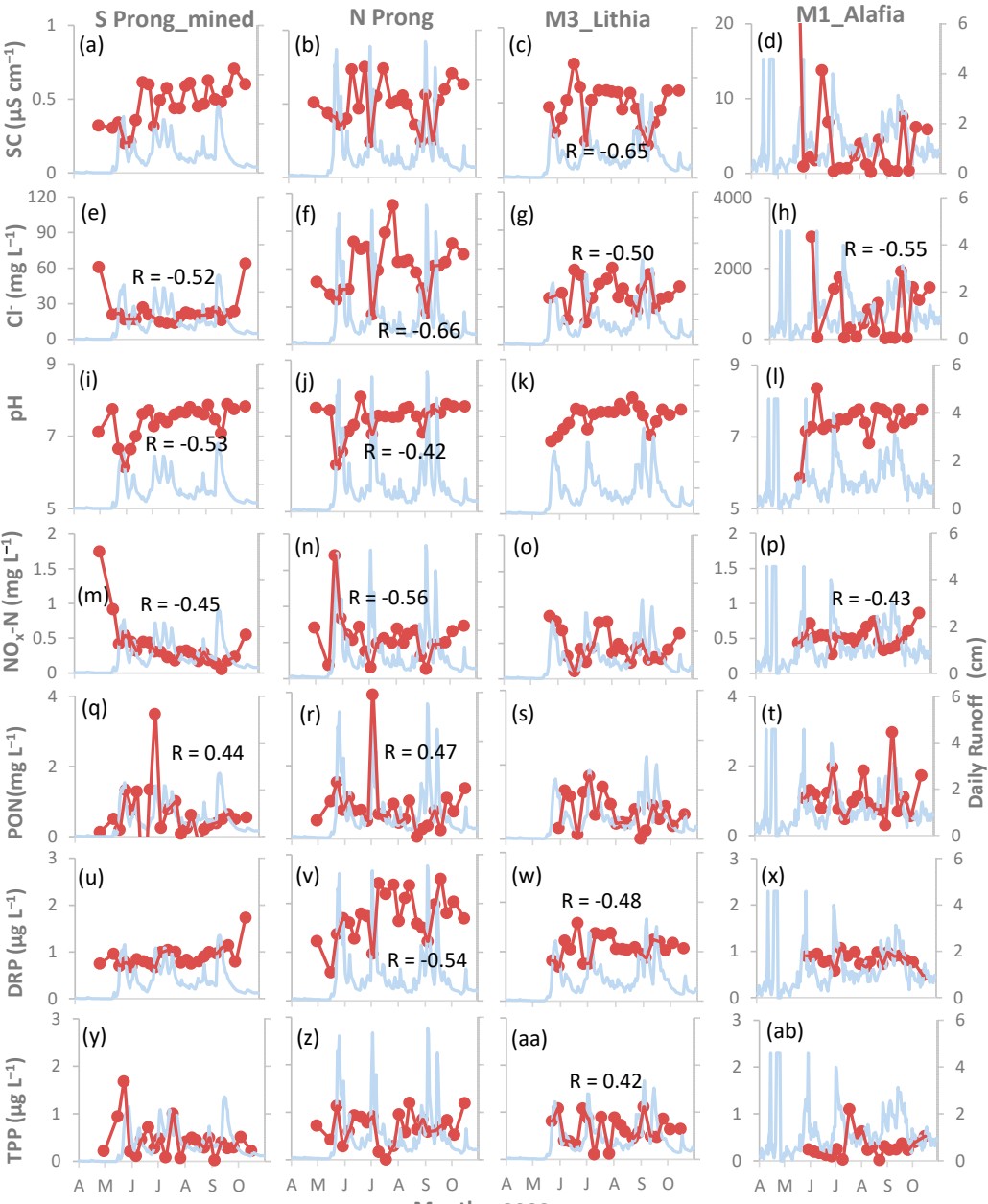

**Figure 6.** Temporal variability (April to October 2009) of (**a–d**) specific conductance (SC), (**e–h**) chloride (Cl⁻), (**i–l**) pH, (**m–p**) NO$_x$–N, (**q–t**) PON, (**u–x**) DRP, and (**y–ab**) TPP in four sub-basins (South Prong, North Prong, mainstem sites M3 at Lithia and M1 at Alafia). Water quality variable scales (red) are marked on the primary *y*-axis, while daily runoff (light blue) is marked on the secondary *y*-axis. A, M, J, J, A, S, and O on the *x*-axis are abbreviations for April, May, June, July, August, September, and October, respectively. Correlation coefficient (R) is included when there was a significant relationship with daily runoff. PON: particulate organic N, DRP: dissolved reactive P, and TPP: total particulate P.

In addition to the above frequent variations, long-term changes in water quality also occurred at the SP—the site with the largest phosphate-mining area. During the wet period (13 May to 20 September), there were decreasing trends of SC and DRP and an increasing trend of NO$_x$–N (Figure 6a,m,u).

Water quality variables and N and P forms at other sites with dominant land use of urban, agricultural, and forest also showed large temporal variabilities (Figures S1 and S2). Owing to the lack of water discharge data for these sites, the relationships with runoff

cannot be quantified. One consistent pattern observed across sites was drastic drop of pH at the beginning of the wet season (Figure S1p–t).

## 4. Discussion

### 4.1. Effect of Phosphate-Mining Reclamation on Stream Hydrology and Nutrient Exports

It is well established that urbanization and agriculture enhance N and P inputs and alter the hydrology, thus accelerating N and P export to receiving waters. The impervious surfaces (rooftops, roadways, parking lots, sidewalks, and driveways) in urban watersheds increase the magnitude of stormwater runoff, forming a typically flashy hydrograph relative to less disturbed watersheds (e.g., [12]). Load duration curve analysis showed that increasing urbanization in watersheds was associated with shifts in N and P export in high-flow conditions [29,30]. The clearing of forested areas in the watersheds for crop and pasture lands and urban sprawl often results in the modification of natural flow regimes and nutrient exports. In N enriched corn-soybean dominated Midwest watersheds, Royer et al. [9] reported that >50% of $NO_3$–N was lost during the high-flow conditions (>90% percentile flow). In Florida, DRP concentrations in the streamwater draining agriculture dominated watershed increased by 10-fold in high-flow events during Hurricane Katrina [30].

Similar to land use/land cover changes caused by urbanization, surface mining alters the hydrology due to the structural changes in the catchment (i.e., topography, drainage density) in addition to changes in the water budget due to loss of forest and soil compaction [31,32]. In this study, however, we found that the runoff in the sub-basin with the most phosphate-mining area (SP) was less flashy (Figure 2) relative to NP sub-basin with less phosphate-mining (more urban and forest land use; Table 1). This could be ascribed to the effect of reclamation that is required for phosphate-mining. After phosphate-mining, overburden, sand tailings, dewatered clays, and debris are returned to the land in various mixes and combinations of layering. Reestablishing wetlands, redeveloping drainage patterns, and reclaiming upland forests are also included in reclamation activities. As a result, surface runoff was less flashy, and TN and TP loads were more evenly distributed in the SP (with more phosphate-mining) relative to the NP (Figure 3) because stream runoff was usually the control variable for nutrient export [29,30].

### 4.2. Effect of Phosphate-Mining on Streamwater P Concentrations and Forms

In the ARW, TP concentrations in the streamwater were 144–277% greater (1.13–1.71 mg $L^{-1}$) than the USEPA numeric criteria (0.49 mg $L^{-1}$) for the region [17]. The concentrations were also one order of magnitude higher than the criteria for the other regions across the USA (range: 0.01–0.05 mg $L^{-1}$) [33] and other studies in the USA (Table 4). Identifying sources and controls on P export from the watershed are critical for nutrient management and water quality improvement in the Tampa Bay estuary. It is known that the transport of P from the human-dominated watersheds (agriculture, urban, mixed) is higher due to the application of fertilizers, manures, industrial effluents, and wastewater discharge [10,34,35]. In our study watershed, it seems that urban land-use had no apparent effect on P concentrations in streamwater because P in the stream draining the highest percent urban land (Buckhorn Creek) was as low as that in the two forested streams (Figure 5). On the other hand, although the two agricultural sub-basins (>22%) had higher P concentrations (Figure 5), P concentrations were not correlated with percent agricultural land use at a significant level (Table 3), suggesting other possible controls such as phosphate-rich geology of the watershed [36].

Evidence supporting that phosphate-mining had a dominant control on P transport included P concentrations (i.e., TP, DRP, and DUP) that were significantly higher in the sub-basins with higher percent phosphate-mining (>10%) than the other sub-basins ($p < 0.05$; Figure 5). Moreover, streams draining a large percent of phosphate-mining land use contributed more P to the river relative to their water contributions (Table 2). Lastly, PRP was positively correlated with the phosphate-mining area, unlike other P forms (Table 3). One

of the possible reasons for uncoupling of DRP or DUP with the phosphate-mining land-use may be due to different type of mining land uses such as active mine lands, reclaimed mined lands, lands owned by mine interests that are yet to be mined, and lands owned by mine interests that cannot be mined [39]. The lower concentrations of TP in the SP, despite 59% phosphate-mining land-use, could be because most of the mined lands are reclaimed in this sub-basin. In contrast, the highest P concentration in the NP (despite 36% mined land-use) was likely due to (1) the presence of active mining, (2) septic leaks due to the highest septic tank density (Table 1), and (3) wastewater discharge (0.35 $m^3$ $s^{-1}$ with 3.1 mg TP $L^{-1}$) from a wastewater treatment plant [40]. Another interesting finding was that DRP and DUP concentrations were positively correlated with watershed wetland land use (Table 3), although wetland accounted for only a small land use (5–14%). The reason for this correlation was not clear, but we know that wetlands can be a source of P to streamwater due to DRP release under organic, anaerobic conditions [41,42]. Another possibility for DRP release from wetlands is that phosphate minerals (e.g., carbonate-fluorapatite) may react with organic acids that are produced from wetlands [43]. The above hypotheses regarding the correlation of streamwater P and wetland warrant further investigation.

**Table 4.** Comparison of concentrations (mg $L^{-1}$) of total phosphorus (TP), dissolved reactive phosphorus (DRP), total nitrogen (TN), and nitrate-nitrogen ($NO_3$–N) between this study and previous studies for the United States.

| Sources | TP | DRP | TN | $NO_3$–N |
|---|---|---|---|---|
| Alafia River (This study) | 1.14 ± 0.05 | 0.74 ± 0.04 | 2.90 ± 0.14 | 0.52 ± 0.03 |
| USEPA criteria this region [17] | 0.49 | | 1.65 | |
| Criteria for other across USA [33] | 0.01–0.05 | | 0.04–0.63 | |
| Indiana, USA (agricultural) [37] | 0.13–0.23 | | 6.9–9.2 | 5.6–7.6 |
| Seattle, USA (urban) [38] | 0.03–0.07 | 0.01–0.04 | 1.1–1.5 | 0.9–1.3 |

Our values are $NO_x$–N, and there might be a minor difference from $NO_3$–N reported in the citations.

Results of SC and pH in this study suggest that they can be used as indicators for phosphate-mining in the ARW. SC is an index of dissolved salts in water, including $Cl^-$ but more importantly, other ions, as SC and $Cl^-$ were not correlated with each other across the sites (Figure 4a,b). The $Cl^-$ was highly correlated with the septic systems density (R = 0.81, $n = 10$, $p < 0.05$), and thus it was an indicator of wastewater input—a point source. That explains why $Cl^-$ was negatively correlated with stream runoff (Figure 6). Because SC was positively correlated with pH, the major components of the ions were likely weak bases that were weathering products of phosphate minerals (carbonate-fluorapatite) and associated carbonate rocks present in the ARW (limestone and dolomite) [44]. That is why both SC and pH were significantly higher in the two phosphate-mining sub-basins. Lower pH values at other sites (especially the forested sites) can be attributed to the loss of acidic organic compounds from the forest floor. More interesting, both SC and DRP were found to increase during the wet season in the SP site regardless of the flow condition (Figure 6). This finding indicates that SC and DRP were probably from the same source—chemical weathering of phosphate minerals. We also can infer that this source was large enough; otherwise, DRP would decline during the wet season.

*4.3. Possible Controls on Streamwater N Concentrations and Forms*

Mean TN concentrations in the streamwater of the ARW (1.99–2.95 mg $L^{-1}$) were much lower than agricultural land use dominated (>60% corn-soybean rotation) watersheds of Indiana, USA (6.2–9.4 mg $L^{-1}$) [37], but were slightly higher than the urban and forest dominated watersheds (1.1–1.5 mg $L^{-1}$) of Seattle, USA [38]. Moreover, the mean TN concentrations at the three mainstem sites were approximately 18–78% greater (1.9–2.9 mg $L^{-1}$) than the USEPA numeric criteria (1.65 mg $L^{-1}$) for the region and the other regions across the USA (Table 4). Therefore, the Alafia River is still a source of N pollution to the Tampa Bay Estuary in Florida, where watershed N management is necessary.

The overall elevated organic N concentrations and dominance of organic N (DON 0.9–1.2 mg $L^{-1}$; PON: 0.5–1.1 mg $L^{-1}$) over inorganic forms (0.3–1.1 mg $L^{-1}$) in the ARW were likely attributed to local climate and ecosystems. Usually, the dominance of organic over inorganic N occurs in forest watersheds [45]. Previous studies have shown that N was primarily lost in the organic forms in the forest and pasture dominated watersheds [8,10,34] due to lower inorganic N inputs and leaching of the organic compounds from the forest covers [46]. In contrast, inorganic N such as $NO_3$–N generally dominated in the developed sub-basins in Baltimore (>80%) and Wisconsin (75%) [8,47]. In the ARW, the higher proportion of DON and PON than $NO_x$–N in the streamwater contradicts the findings of previous studies [8,47]. This difference can be ascribed to highly productive ecosystems coupled with Florida's warm weather [48]. In these ecosystems, $NO_3$ is generally transformed into organic N by vegetation and microbes [49], while organic N is leaked from the ecosystems to streamwater [50] due to factors such as lack of P limitation, warm temperature, and high moisture content in the soils [7,51,52].

Despite the dominance of organic N, we also observed a larger variability of $NO_x$–N (followed by PON) over other forms (Figure 5), which was positively correlated with percent watershed agricultural land use (Table 3). It is well known that agricultural lands receive far higher anthropogenic N inputs (fertilizers and manures) than forest and even than urban land use (industrial effluents and wastewater discharge) [8,10,34]. The lowest TN and $NO_x$–N concentrations in the forested sub-basins (Figure 5) can be due to lower anthropogenic N inputs [10,34,35], as well as more N uptake by plants and denitrification by microbes [8,34].

### 4.4. Molar N:P Ratio and Implications for Nutrient Management

We determined the molar ratio of dissolved inorganic N (DIN = $NO_3$–N + $NH_4$–N) and DRP in different sub-basins (Figure 5), which ranged between 0.8:1 and 7.0:1. These values were lower than Redfield Ratio i.e., DIN:DRP = 16:1 [53]. This suggests that the growth of algae in the receiving water (e.g., the Tampa Bay) is more limited by N than P [3,54]. These low DIN:DRP ratios were more attributed to elevated P inputs from the upper part of the ARW, especially from those sub-basins that drain the phosphate-mining areas because of the larger variability of P than N.

Since N is the limiting nutrient for algae growth, further studies should focus on two key areas of N watershed management, including identifying the sources of organic N, its bioavailability, as well as potential biogeochemical and hydrological controls. In the developed sub-basins, stormwater retention ponds and septic systems are potential DON sources [53,55]. In agricultural and urbanizing watersheds, N and/or O stable isotopes can be a useful tool to identify the dominant sources of N, e.g., [52]. This could further provide a way forward for conceptualization N sources in developed watersheds. Even so, the sources of P and watershed P management cannot be ignored because of the elevated high P concentrations in the Alafia River. Because sub-basins of the upper watershed with a large phosphate-mining area were high in P concentrations and loads, the efforts in the upper watershed need to focus on P reductions. The role of wetlands, which was positively correlated with P concentrations, in P loss warrants further investigation.

### 5. Conclusions

Our results showed that the sub-basin with the highest phosphate-mining area had less flashy flow peaks and a more evenly distribution of N and P loads relative to an adjacent phosphate-mining sub-basin. Therefore, the current reclamation of phosphate-mining in the ARW can restore stream hydrology and nutrient exports. The ARW had 18–78% greater TN (1.9 to 2.9 mg $L^{-1}$) and 144–277% greater TP (1.1–1.7 mg $L^{-1}$) than USEPA numeric nutrient criteria threshold values for the region. Concentrations of TN and $NO_x$–N were higher in the agricultural sub-basin and positively correlated with percent agricultural land use, suggesting an abundance of N sources in the agricultural sub-basins. Organic N forms (DON, PON) dominated the streamwater of all sub-basins owing to highly productive

ecosystems coupled with Florida's warm climate and year around vegetation. On the other hand, phosphate-mining sub-basins had significantly higher concentrations of TP and dissolved P forms than sub-basins without mining influence. Further, these sub-basins contributed more P to the Alafia River than the water discharge—all indicating influence of phosphate-mining on nutrient export. Meanwhile, the pH and SC were positively correlated with percent phosphate-mining area, and inorganic P forms (DRP, PRP) were dominant in all sub-basins (in contrast to N), suggesting P source from phosphate weathering in the watershed. Based on the molar inorganic N to P ratios, we conclude that our study watershed is more N than P limited, suggesting best management practices should focus on reducing N pollution and improving reclamation of phosphate-mining areas to reduce P inputs.

**Supplementary Materials:** The following are available online at https://www.mdpi.com/article/10.3390/w13081064/s1, Figure S1: Temporal variability of specific conductance, chloride ion, dissolved oxygen, and pH across sub-basins with dominant land use of urban, agriculture, and forest, Figure S2: Temporal variability of nitrogen and phosphorus forms across sub-basins with dominant land use of urban, agriculture and forest.

**Author Contributions:** Conceptualization, G.S.T.; methodology, K.B. and G.S.T.; validation, S.D.; formal analysis, S.D.; investigation, K.B.; resources, G.S.T.; data curation, S.D.; writing—original draft preparation, S.D. and K.B.; writing—review and editing, S.D. and G.S.T.; visualization, S.D.; supervision, G.S.T.; project administration, G.S.T.; funding acquisition, G.S.T. All authors have read and agreed to the published version of the manuscript.

**Funding:** This research was funded by the former Soil and Water Quality Laboratory at the Gulf Coast Research as well as a graduate research assistantship to Kamaljit Banger from the Soil and Water Sciences Department of the University of Florida. Gurpal S. Toor received partial funding from USDA Hatch project 1014496 and USDA-NIFA AFRI competitive grant 2018-09093.

**Institutional Review Board Statement:** Not applicable.

**Informed Consent Statement:** Not applicable.

**Data Availability Statement:** All data that support the findings of this study are included within the article and supplementary materials.

**Acknowledgments:** We extend special thanks to Stefan Kalev, former MS student, for his help with drawing Figure 1.

**Conflicts of Interest:** The authors declare no conflict of interest.

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
