# Peer review of "Evidence of Phosphate Mining and Agriculture Influence on Concentrations, Forms, and Ratios of Nitrogen and Phosphorus in a Florida River"

_water, doi:10.3390/w13081064_

Round 1

Reviewer 1 Report

This is an important manuscript on phosphate-mining affects nutrient exports to coastal waters, the study suggest that phosphate-mining and agriculture are important sources of streamwater P and N, respectively. It presents important data on the transport of inorganic and organic N and P forms from diverse land use in a watershed, especially help design and fine-tune effective nutrient pollution remediation programs for coastal waters. In general, the results are well presented and the conclusion has useful implication for understanding P inputs from mining alter N:P ratios in receiving waters.Thus, this paper could be accepted.

Author Response

Co-author Dr. Toor, and Shuiwang Duan have carefully read through the paper. We have corrected spelling and grammar errors, and believe that English of the paper has greatly improved.

We've also checked every tables and figures and improved their quality.

Reviewer 2 Report

The submitted manuscript elucidates the effect of phosphate mining and agriculture on nutrient dynamics in Alafia River Watershed, Florida. Although the impact of agricultural activities on nutrient loading and watershed hydrology has been well studied, not that much information is available regarding the effect of phosphate mining on nutrient dynamics. The study presented in this manuscript therefore includes interesting findings. However, there is a huge gap (more than a decade) between the year of research and the publication of the results. It would be more significant if the authors could publish the results earlier as the non-point source pollution is generally episodic and influenced by spatial, meteorological, and geological factors. The reviewer would like to know the reason for the delayed disclosure of results. In addition, there are several other issues that need to be clarified or improved before its further processing. Please check the following comments:

  1. L33: Should be ‘suggests’
  2. Table 1: Please include the source of information.
  3. L116: Should be ‘4 °C’
  4. Figure 2: It would be better to indicate the 23 sampling events in the figure and just write the title of the figure. No need to include data source, calculation method along with explanation in the title.
  5. L136: EPA Method 365.1, this method is not developed by the authors in the referred article. Please cite the original reference.
  6. Section 2.4: Please cite the links as references.
  7. L153: Please explain NOAA.
  8. Section 3: It would be better to include another sub-section here showing the summary of the sampled rain events or characteristics of the rain events.
  9. L187-188: Please correct the figure numbers.
  10. Figure 4: (L216) SC is already mentioned in L214.
  11. Section 3.3: Please recheck and correct the figure numbers.
  12. Please explain the acronyms at the time of their first use and then continue using the acronyms. Check and correct such issues throughout the manuscript.
  13. For a robust discussion, it would be better to include a table comparing the results of this study with other regions.

The author mentioned in the Materials and Methods, they did a posthoc test to show the differences in nutrient forms in streamwater among the sub-basins. However, there is no indication of such in the figures.

Author Response

1. L33: Should be ‘suggests’

Response: We have this sentence to "We suggest that.." (line 33).

2. Table 1: Please include the source of information.

Response: We have added the sources of information in the notes of the Table (line 105-107).

3. L116: Should be ‘4 °C’

Response: We have corrected “4oC” to “4 °C”. (line 121).

4. Figure 2: It would be better to indicate the 23 sampling events in the figure and just write the title of the figure. No need to include data source, calculation method along with explanation in the title.

Response:  We have now added 1-23 to the top of the figure to indicate the 23 sampling events. The information on data source and calculation have been moved method to Method Section, and the explanation in the title was moved to Results Section.

5. L136: EPA Method 365.1, this method is not developed by the authors in the referred article. Please cite the original reference.

Response: We have now cited the original reference (citation 24; O'Dell, J. W. 1993) for this method (line 133).

6. Section 2.4: Please cite the links as references.

Response: We have moved the links to the reference list and cited them as references. (line 151, 154).

7. L153: Please explain NOAA.

Response: We have now explained NOAA as National Oceanic and Atmospheric Administration (line 150).

8. Section 3: It would be better to include another sub-section here showing the summary of the sampled rain events or characteristics of the rain events.

Response: We have now broken down the previous sub-section 3.1 into two sub-sections (3.1 and 3.2) (Line 169-198). Sub-section 3.1 shows the summary of the sampled rain events or characteristics of the rain events.

9. L187-188: Please correct the figure numbers.

Response: we have corrected the previous wrong figure numbers “Fig. 4b and 4c” to “Fig. 3b and 3c”.

10. Figure 4: (L216) SC is already mentioned in L214.

Response: We have deleted the unnecessary SC explanation from the caption of Figure 4.

11. Section 3.3: Please recheck and correct the figure numbers.

Response: we have corrected the previous wrong figure numbers “Fig. 6b and 6c” “Fig. 6b” to “Fig. 5b and 5c” and “Fig. 5b”.

12. Please explain the acronyms at the time of their first use and then continue using the acronyms. Check and correct such issues throughout the manuscript.

Response: We have checked acronyms throughout the manuscript and explained them at the first time of use and then continue to use it. The acronyms include, FL, NOAA, N, P, ARW, SC, DO, M1, M3, Cl, NP, SP and so on.

13. For a robust discussion, it would be better to include a table comparing the results of this study with other regions.

Response: we have included a table (Table 4) comparing the results of this study with other regions (line 339-342).

14. The author mentioned in the Materials and Methods, they did a posthoc test to show the differences in nutrient forms in streamwater among the sub-basins. However, there is no indication of such in the figures.

Response: we have now added the results of posthoc test in Fig. 4 and Fig. 5.

BTW, co-author Dr. Toor, and Shuiwang Duan have carefully read through the paper. We have corrected spelling and grammar errors, and believe that English of the paper has greatly improved. We've also checked every tables and figures and improved their quality.

Reviewer 3 Report

The article is original, innovative, presented in an understandable way,
However, a few observations:
1. The laconic title of Figure 2 is required. A description of the image is now provided.
2. 3.1. Chapter: lines 169 to 173 should be in the methodology section as a discussion of Figure 2.
3. Figure 6 is not very clear. What is marked in red, what is marked in blue? The x-axis must also be explained.

Author Response

1. The laconic title of Figure 2 is required. A description of the image is now provided.

Response: We have moved the information on data source and calculation method to Method Section, and the explanation in the title was moved to Results Section. Now, the title is now laconic.

2. 3.1. Chapter: lines 169 to 173 should be in the methodology section as a discussion of Figure 2.

Response: This paragraph describes relationship between Rainfall and Surface Runoff and difference in hydrology between sub-watersheds.  It is part of our observation and highly related  to nutrient export. So, we keep it here as section 3.1, and put the rest of the section as 3.2.

3. Figure 6 is not very clear. What is marked in red, what is marked in blue? The x-axis must also be explained.

Response: we have added one more sentence in the caption of Figure 6 to explain the graph in red and in blue: “Water quality variable scales (red) are marked on the primary y-axis, while daily runoff (light blue) is marked on the secondary y-axis.” (see line 295-296).

Co-author Dr. Toor, and Shuiwang Duan have carefully read through the paper. We have corrected spelling and grammar errors, and believe that English of the paper has greatly improved. We've also checked every tables and figures and improved their quality.

Round 2

Reviewer 2 Report

The reviewer appreciates the authors efforts to improve the quality of the manuscript. After the revision, the manuscript as a whole is more logically described.